# Blockchain Adoption in Academia: Promises and Challenges

**Artyom Kosmarski**

Laboratory for the Study of Blockchain in Education and Science (LIBON), State Academic University for the Humanities (GAUGN), 119049 Moscow, Russia; kosmarski@gaugn.ru

**Abstract:** Blockchain has received considerable attention recently, due to its promises of verifiable, permanent, and decentralized data handling. In 2017–2020, blockchain (and associated technologies such as smart contracts) has progressed beyond cryptocurrencies, and has been hailed as a disruptive technology for a score of industries. This study adds to the growing body of research on blockchain adoption and blockchain-driven innovation in various fields, including transport, finance, and education. However, the impact of distributed ledger technologies (DLT) on the management of science has not been systematically studied so far. This paper aims to fill the gap by studying the experience of adoption of blockchain-based solutions in academia in 2017–2020. The research is based upon a critical review of projects, relevant literature, and qualitative research: interviews ($N = 24$) and focus groups ($N = 4$) with startup founders, scholars, university executives, librarians, and IT experts from the European Union (EU), the United States of America (USA), Russia, and Belarus. Key challenges and barriers to blockchain adoption in academia are delineated: usability and security issues, legal concerns, conflict of values, and a critique of political dimensions of blockchain governance.

**Keywords:** blockchain technology; distributed ledger; science; governance; adoption

## 1. Introduction

The rapid development of blockchain (distributed ledger technology, DLT), building upon the success of Bitcoin, has made it a promising and potentially disrupting technology in many areas. In 2017–2018 blockchain (and associated technologies, such as smart contracts) has been adopted in banking, retail, supply chain management, healthcare, even public administration [1–3].

In a nutshell, a blockchain is a set of data blocks connected by cryptographic tools, in order to make it impossible to change the content of one block without interfering with all the others. In that digital ledger, information is stored in a network of decentralized nodes, and all recorded transactions are transparent to each member of the network. This approach towards handling data (decentralized and distributed) prevents retroactive altering of data, e.g., for fraudulent ends. Essentially, blockchains enact a consensus mechanism that ensures the accuracy of a transaction without the necessity to trust transacting parties [4]. Distributed ledger technology is an umbrella term for databases that employ independent nodes in order to record and share data in a decentralized network, with blockchain being just one of them (using cryptographic tools to link blocks of data, various consensus mechanisms). Still, in this paper, the terms blockchain and DLT are used interchangeably.

The appeal of blockchain to industry and academia builds upon the promise to make data reliable, immutable, transparent, and decentralized. However, it is not the data handling, but the social appeal of blockchain [5] that has attracted the attention of academia. Principal advantages of blockchain in this perspective, apart from the immutability and verifiability of data, is the guarantee of trust in the trustless environment and successful peer-to-peer interactions without the need for a central governing

body ("the third party"). These features dovetail with the logic of modern science: it is international, decentralized—there is no governing body that decides everything [6]—and develops thanks to networks of trust within the academic community (peer review system and invisible colleges) [7]. The analogy was not lost on a few early observers: "Scientific information in its essence is a large, dynamic body of information and data that is collaboratively created, altered, used and shared, which lends itself perfectly to the blockchain technology" [8] (p. 8).

However, the impact of blockchain on the management of science has not been systematically studied so far—a gap that this paper attempts to fill. The need for such research is justified by the fact that quite a few DLT-based projects in science have been launched in 2017–2019 (for a recent review see [9] and Section 3 below), and those have been operating by trial and error, frequently failing in their attempts to gain acceptance among both scientists and university/lab executives. A systematic study of key obstacles and challenges that are encountered by blockchain projects in science could make the future trajectory of DLT adoption (and open innovation in science, more generally) straighter and faster. There is a handful of studies of blockchain technologies for open science, but they have either focused on prospects and promises of this new technology (e.g., [8,10]) or based on extensive literature review and not on original research [11].

This paucity of research on DLT implementation in science stands in contrast to the growing body of research on blockchain adoption in various fields, including transport, finance, and education. There is an ongoing investigation of factors limiting the potential of DLT to revolutionize industries and make good its promises: factors including, but not limited to, the lack of understanding of blockchain technology, cultural resistance, regulatory practices (need for new laws), governance issues (who owns and governs the DLT network), the legal validity of smart contracts, and technical challenges, i.e., scalability and security [12–16]. Moreover, some studies have focused on blockchain as a vehicle of innovation, e.g., economic governance [17], international trade and taxation [18], and a consensus-driven replacement for bureaucracy [19].

Recently, a few papers have moved beyond that and made a conceptual revision of blockchain proponents' ambitions, with a critique of key concepts (e.g., trust [20]) and examination of novel use cases within a state-business-community nexus [21]. Still, a plurality of papers on blockchain and related technologies is structured aseither a proposal (what blockchain could do) or a critical literature review [16]. There is a lack of studies employing qualitative methodology [22] and probing practical experience of DLT adoption, the lessons learnt by those "on the ground" (a notable exception is a study of Kenyan IT-entrepreneurs [23]).

This paper aims to fill the gap: its aim is to gauge the experience of adoption of blockchain-based solutions in academia in 2017–2020. It is arguably the first study of DLT use in science made within a framework of blockchain adoption studies, and one that attempts to evaluate the experience from the actors' point of view, drawing on the qualitative methodology. The research is based upon a critical review of projects, relevant literature, a set of interviews ($N = 24$), and focus groups ($N = 4$) with startup founders, scholars, university executives, librarians, and IT experts from the European Union (EU), the United States of America (USA), Russia, and Belarus. In the following sections, I would give an outline of blockchain applications in science, and delineate key challenges and barriers to DLT adoption in academia, elicited via the interviews and focus groups (The whole issue of blockchain in higher education (academic degree management, evaluation of learning outcomes, etc.) has been deliberately left outside of the scope of this paper. There is substantial literature on this topic, e.g., [24–26]).

The rest of the paper is structured as follows: section two describes the materials and methods of this research. Subsequently, in section three, I present an overview of blockchain-based solutions for the woes of academia (research data verification, academic publishing, and peer review). It is followed by the results section (key themes of DLT adoption in academia, from technical difficulties to political issues) and the discussion, where the key trends for the future are laid out.

## 2. Materials and Methods

The aim of the research was to collect and analyze data on agents' perception of DLT implementation in academia, and then draw meaningful insights from their narratives—insights that would reveal key challenges to DLT adoption in this area. The analysis is based on 24 in-depth semi-structured interviews and four focus groups conducted by the author and his collaborators in November 2018–July 2020 (see Appendix A). The participants were recruited via e-mail and social networks (the objective was to reach a varied sample of experts that have had an experience of organizing or evaluating DLT projects for science). All of the subjects gave their informed consent for inclusion before they participated in the study. The study was conducted in accordance with the Declaration of Helsinki.

I employed purposeful sampling [27] in order to achieve a balanced sample, to access the perspectives of both DLT evangelists, startup founders, and experts (*N* = 11), on the one hand, and university professors, deans, and staff (*N* = 13), on the other—the more realistic and conservative side, in a sense. I made efforts to communicate with the representatives of institutions that are visible in the field of blockchain studies (e.g., Research Institute for Cryptoeconomics at the University of Vienna, Blockchain & Society Policy Research Lab, at the University of Amsterdam, TechnischeInformationsbibliotek in Hannover, Germany) and startups that have already launched a functional product actually in use by scientists (e.g., DEIP, ARTiFACTS, bloxberg, Melda.io). Although the sample is not too big, the author assumes that theoretical saturation [28] was achieved, as no substantially novel facts were elicited by final interviews.

The interviews were conducted in person or by Skype, and they lasted, on average, 60 min. Focus groups (group discussions of a set of particular topics) were conducted in three universities, with professors, staff, and post-graduate students who have demonstrated awareness and interest in the issues of blockchain. The focus groups lasted 90 min. The interviewees were selected from a broad range of institutional and geographic environments (USA, United Kingdom, Ireland, The Netherlands, Germany, Austria, Belarus, and Russia), yet the focus groups were conducted in the author's country of residence (Russia). However, this fact should not be seen as an obstacle towards generalization of this study's findings: Russian universities face the same challenges as their counterparts across the globe (managerial governance, competitiveness, push to innovate [29,30]); besides, Russia has a developed IT economy [31] and it does not substantially lag behind Europe in terms of DLT innovations.

The interviews and focus group discussions were structured around the following list of questions, prompting a spontaneous discussion of the topics that are covered by the guide:

- How would you define blockchain and smart-contracts? What fundamental problems are these technologies supposed to solve? Where they could be implemented?
- Could you think of an example of a successful application of blockchain in academia? In what field DLT might be useful—data storage, intellectual property, publishing, novel funding instruments, anything else? Any thoughts about decentralized autonomous organizations (DAOs) in science?
- Where and to what extent DLT technologies have been used by researchers, executives, staff, and Ph.D. students in your institution /your country? What barriers and challenges have they encountered? Do they fit into regulatory/legal framework? If you think your environment is not yet ready for these technologies, why is that so? What would change in 1–2–5 years' time?

Interviews and focus group discussions were transcribed, and then subjected to a coding procedure: starting from elucidating a set of themes towards mapping and interpretation of the whole body of data through the lens of those themes. Data analysis was also based on the methodology that was outlined by [32]: a three-step process, leading from categorization of responses (1) towards second-order themes (2) and, finally, clustering categories into aggregate dimensions (3). Transcripts would be available from the author at request.

## 3. Blockchain Applications in the Management of Science

Blockchain and crypto projects were slow to emerge in this field, as science is not strictly business and still depends on public funding (unlike fintech, for instance). The first tentative projects and

academic publications appeared in 2016 (e.g., a paper on improving data transparency in clinical trials using blockchain smart contracts [33]). Subsequently, at the peak of the ICO boom in 2017, several startups promised to solve all of the problems of science with their tokenized economy, with some of them ending up as scam (globex.sci, scientificcoin.com), others just petering out for the lack of funds (for a full list of crypto projects in science and their up-to-date status see [9]). Eventually, blockchain in academia proved more workable as a set of solutions for specific processes (peer review, data storage, funding, etc.).

At the same time, there has emerged an extensive network of scholars, IT-experts, and crypto-enthusiasts, many of whom have joined Blockchain for Science association (blockchainforscience.com) and research groups in Europe and the USA. Finally, corporations began to enter this field—IBM, for instance, has obtained a patent for a platform for the collection and analysis of scientific data on the blockchain [34]. Still, DLT projects in academia have presented themselves (without exception) as a solution to a certain problem of how science is organized and practiced.

### 3.1. Open Data

There are issues at all stages of the research cycle in science, from dubious data collection and protocol procedures to the distorting of evidence to fit the hypothesis [35], as well as p-hacking [36]. The pressure to publish and journals' preference for positive rather than negative results (positive-results bias) have led to the formation of the so-called false chain of research: new studies are based on untested (and probably erroneous) old ones. However, these problems are not exclusively caused by the malicious intent of fraudulent scientists. There are many reasons for false data, from inevitable inaccuracies in large collaborations to salami-slicing (need to publish even small results as soon as possible to extend funding [37].

Blockchain solutions promise to fix that by making the research cycle open and transparent, and by facilitating data sharing. Discoveries may be rapidly recorded in the distributed registry, which indicates the authorship and date of discovery. Time-stamping on the blockchain could be a novel way to protect ideas without resorting to (slower) patents and publications [38,39]. Furthermore, blockchain allows for tracking the entire scientific project from hypothesis to data collection and further analysis. The stability of data on the blockchain is essential here. By uploading the data into DLT and making it open to a broader academic community, researchers would no longer be able to tamper with it, to rig the data, to remove outliers, etc. [11,40] (pp. 15–17). As of 2020, arguably the most advanced projects of this kind are US-based ARTiFACTS platform (it uses blockchain technologies to create an immutable record of research material, allowing for individuals and groups to register material and secure its provenance while providing public access) and bloxberg—a global blockchain network that is run by a consortium of universities and research organizations (it provides data certification system and a structure to develop decentralized applications for scientists).

### 3.2. Publishing and Peer Review

Academic publishing is passing through turbulent times—the crisis of the subscription model, the rise of open access journals, and potentially disrupting Plan S with its forceful drive of compulsory transition to open access journals [41]. The publishing cycle in academia is extremely slow, regardless of the payment model: writing an article, submitting it to a journal, searching for reviewers, getting feedback, and finalizing takes months if not years [42]. Finally, reviewers are increasingly overworked and underpaid [43]. The need for incentives in peer review, e.g., "academic dollars" was voiced well before the advent of cryptocurrencies [44].

Many projects discussed the advantages of DLT for solving these problems. Minimally, it provides notarization. Recording a text or even a draft idea on the blockchain (time-stamping) allows for a scientist to assert priority and intellectual property rights, and then he or she might freely share it as a preprint, thus speeding up the exchange of ideas [45].

Finally, the decentralization and disintermediation principles that are behind blockchain suggest an independent publishing platform where authors and reviewers interact directly with each other in a p2p network, with no need for excessive publishing and subscription costs (while the reviewers are rewarded with tokens. A token, in crypto economy, is a unit of value programmed and operated on any blockchain. There is another definition of a token, more geared towards its commercial properties: "a unit of value that an organization creates to self-govern its business model, and empower its users to interact with its products, while facilitating the distribution and sharing of rewards and benefits to all of its stakeholders" [46]. Not surprisingly, this idea has been so appealing that virtually each blockchain startup in science promised an open access platform (scienceroot.com, eurekatoken.io, pluto.network, orvium.io). Apart from mere repositories, they offer reputation and incentives systems, as well as mechanisms for research management and collaboration. Many of these over-ambitious platforms have failed to reach the MVP stage or they have been aborted (Orvium is a notable exception).

### 3.3. Research Funding and Incentives for Scholars

Scientists have to spend a great deal of their time writing reports, grant applications, and doing paperwork, all in order to get their research funded [47,48]. Finally, funding for science has been shrinking. Governments are gradually moving away from large-scale research funding, hoping that business, industry, and private foundations will replace it [49].

How blockchain could fix these issues? First of all, an automated system of disbursement of funds with transactions on smart contracts could reduce the overhead costs and ease the burden on accountants, auditors, and scientists themselves. It would save them from filling out a lot of papers and make the whole process of allocation and distribution of funds more efficient. Additionally, a funder might set a combination of conditions (e.g., citations, articles, datasets) and peg the grant money to the fulfillment of these conditions (through smart contracts)—this approach is implemented in the DEIP blockchain ecosystem (deip.world). Another successful example—in the public sector—is the pilot project of National Research Council Canada: prototype where Ethereum blockchain was employed to proactively publish grants and contribution data in real time (https://nrc-cnrc.explorecatena.com/en).

However, the innovative drive of DLT lies in the fact that science funding itself, as well as incentives for scholars could be changed, "*We can experiment with new money distribution schemes, grant schemes, and that would bring about cultural change. With blockchain, things would change much quicker*" (interviewee 4). Entering the cryptocurrency sphere gives scientists a chance to find money from investors whose interests and outlook are very different from universities. In such cases, DLT framework would provide investors with a guarantee against scams and roguish projects: all of the initial data and development of the research can be traced, and the allocation of funds can be pegged to the achievement of certain milestones (Molecule Catalyst, a reward-based, crowdfunding platform for basic research in drug development, is arguably the most advanced project of this kind, with payments handled by the Ethereum network, in a decentralised way, and bonding curves [50] employed as a tool forautomating the distribution of tokens in exchange for contributions).

Moreover, one could receive tokens if their research results are validated independently by others, or used in their future work, as an economic equivalent of citation. Thus, token economy creates a potentially powerful channel for financing and implementing breakthrough ideas, even in basic science [40] (p. 25).

In this sense, crypto-economic tools could make scholars more independent, by opening an alternative channel of recognition and funding, providing scientists with a clear economic interest in order to engage in the crypto economy. In the end, science would get more independent economic agents, apart from the state and big funding agencies and philanthropies, "*There we have more opportunities for more independent players, and for more intermediaries—and that is an interesting contradiction, as they say that blockchain is all about disintermediation. And we will probably have more players here, with less bureaucracy and more efficient, blockchain-based ways to redirect money in different directions*" (interviewee 9).

*3.4. Decentralized Governance and Token Economy*

Apart from a set of technical solutions, blockchain for science projects in 2017–2019 have moved forward with a more ambitious goal of disrupting science as a whole [40] (pp. 19–22). The impetus of these projects is mounting dissatisfaction with the oligopoly of large publishing houses, the "tyranny of metrics" [51] and indicators, the alarming growth of biased and non-reproducible research, and the "precariatization" of scientists. Blockchain addresses these concerns with its promise to radically restructure the rules of the game in academia:

- to foster open science, transparent transactions, and decision-making;
- to set up new communities based on common rules set in code [52]; and,
- to give an opportunity for scientists themselves to determine what is important (for example, to encourage reproducible research) Section 5.3.2 of [11].

Accordingly, the idea was not just to introduce token economy and funding from peers [53] into the everyday life of scientists [54], but to restructure the governance of academia. Here, the inspiring concept was the DAO, decentralized autonomous organization): a programmed set of rules, which are transparent, controlled by users, and not influenced by a central authority [55]. In other words, DAO functions as a new form of scalable, self-organizing cooperation, and operated by smart contracts on the blockchain.

DAOs could free individual scientists and research collectives from red tape, regulatory pressure, forced competitiveness, and allow for establishing global horizontal networks to collaborate on doing ground-breaking research, in "independent and self-coordinated initiatives" [6]. Subsequently, "private and governmental funders as well as individuals may invest in "Science DAOs" and foster the creation of great research collaborations that can scale globally. These massive research organizations may not be operated by a single hierarchical administrative layer, but through a distributed network of financial and intellectual contributors" [56].

## 4. Results

In the previous section, I provided an overview of contemporary DLT-based projects in science/academia, their goals, solutions, and ambitions. I will now proceed by analyzing interviewees' narratives and experiences of DLT projects in their academic environments. It is worth pointing out that their attitude is generally critical/skeptical: they do not deny blockchain out of hand (i.e., stressing that there is absolutely no need for DLT solutions in their activities), but rather pinpoint trouble spots of blockchain-based projects and try to figure out potential solutions. The responses have been grouped and aggregated under six key themes, as presented in the following subsections.

*4.1. Perceived Immaturity and Weak UX/UI of Blockchain Applications*

The informants' first concern is that blockchain applications for academia at the current stage are not simple and user-friendly enough. Adoption on the part of scientists cannot go on, unless they reach the status of Google Documents or Telegram—easy to download, easy to use, satisfying users' wants, "*Well, until it's as simple as Facebook or Uber, it would hardly achieve success. We set up our platform [a registry for patents and other intellectual property objects on Hyperledger, used by a consortium of Russian universities], and everybody groans. Senior professors are not enough tech-savvy to use it, and young postdocs complain that two clicks are not enough to download an object—as they do in ordinary file sharing sites*" (interviewee 16).

This challenge is to another barrier: university executives have been quick to understand that to move beyond hype [26] andintegrate blockchain into the workflow of their institutions would require tremendous resources, whereas the benefits of DLT are not quite clear. Recording vast amounts of research data on the blockchain requires considerable computation power, and for most blockchains currently in use it would be a long and expensive process, "*To proclaim that our university would be the first to introduce blockchain is no rocket science. But what then? Not every principal can grasp what blockchain is. And those who did understand, they were absolutely aware of the costs of implementation, in terms of money*

*and human resources. The technology itself is not freely distributed, and one needs to hire a score of software developers, what for*?" (interviewee 22).

## 4.2. Institutional Inertia. Regulatory and Legal Issues

Despite the fact that blockchain projects offer publishing solutions, based on open science principles (easy and fast peer review, no intermediaries between authors, no subscription costs), the success of these platforms has been limited so far, with less than a hundred papers in each, "*Who needs an open science journal on blockchain when there are plenty of good journals in my field? They are already indexed in Scopus/Web of Science and have a good reputation*" (interviewee 15) This trend is well-known in the sociology of innovation [57]: the revolutionary benefits of new technology are insufficient to draw users away from customary practices.

Most scientists are not entrepreneurs, and they work within complex institutional structures, while the latter are not always friendly towards new technologies. The efficiency of blockchain solutions could force them to make tough choices in labour policies, "*Smart contracts might efficiently indicate expenses of the lab—test animals, chemical agents, etc.—and allocate necessary funds. But what if you have an accountants office working with these expenditures, for many years, 50 people in all, who would think of firing them?*" (interviewee 18).

Furthermore, the dubious reputation of crypto-economic tools (ICO scams, hacker attacks targeting crypto wallets), sluggish transaction speeds of many blockchains makes university authorities doubt the practicality of introducing DLT into their finances [58,59]. Finally, in many countries, smart contracts and cryptocurrencies are still in legal limbo, further constraining the use of these technologies in academia, "*Cryptocurrencies may not be used as legal tender, therefore we cannot implement DLT technologies in public sector. Every time we have to purchase gas on Ethereum for our smart contracts, we face the risk of prosecution*" (interviewee 18); "*Alright, we set up a platform, a registry of intellectual property objects on the blockchain. University professors and staff can upload and register their discoveries, inventions, designs. But they can neither trade nor receive royalties, the final, financial layer is missing! And that is why the users have little incentive to spend their time on using the platform. Okay, we registered, then what?*" (interviewee 16).

## 4.3. Blockchain Solutions Require Universal Adoption

The afore-mentioned challenges and barriers are not unique to blockchain in academia, and they have been noted by the scholars studying DLT adoption in other areas [60]. However, there exist more peculiar challenges. Virtually every blockchain for science project praises the potential of this technology to encourage open, fraud-free science, where all stages of the research cycle, from data collection to publishing, are transparent. However, transparency and availability of research data on DLT require mass support of scientists. If only a handful of enthusiasts pursues this task, it will not become the norm for the academic community, and the ambitious project would fail: "*We set up a blockchain platform for the journals published at our university. We call it, among ourselves, «Honest publisher». Everything is open and can't be rigged: when an article was submitted, the reviews, editors' decisions. But it's more like a toy, nobody actually wants to use it. Such projects succeed only when a majority of journals support them. And journals in this country have no desire to disclose their dubious practices—fake reviews, co-authors added at the last moment in exchange for bribes, etc. Only when the biggest and respected universities in the country endorse such a platform, endorse blockchain-enabled transparency, and set an example, only then it would work*" (interviewee 1).

Speaking of DAOs and new governance on the blockchain, the informants stress that they face vigorous time constraints and they are naturally reluctant to engage in community projects that demand active participation, discussion, voting, and other obligations of participatory democracy [61]: "*The state is motivated to employ blockchain to investigate research fraud. The state, the university, not the scientists themselves. We planned to structure the whole process of evaluating publications and dissertations in our university on the blockchain. But the project failed because nobody was motivated. Everybody cares only about her data, and not about checking and evaluating her colleagues' input*" (interviewee 17).

Making all of the research data transparent and verifiable takes researchers' time without bringing apparent benefits in their careers. Therefore, open science on the blockchain could probably be implemented only from above, forced upon the scientists by public agencies and foundations. However, this fact runs hard against the grain of decentralization/e-democracy ideology of blockchain projects, and that is grudgingly acknowledged even by blockchain enthusiasts: "*The state must fund blockchain and related digital infrastructure, there is no other option. It is a difficult and complex task. You can't do these things through short-term grants or private money*" (interviewee 12). Individual researchers lack resources and initiative to "blockchainify" their workflow. Moreover, COVID-19 pandemic and the urgent need to restructure workflows, in order to cope with working and learning from home have dealt a serious blow to blockchain-based projects in academia, as the latter frequently require serious investments on the part of universities and relevant public agencies. "*Then [in 2019] quite a few pilot projects started [academic degree records and certifications management on blockchain]. Our university, then some Siberian universities. And then COVID-19 happened, and everybody have had more important issues to attend to. We don't know whether the projects would ever be unfreezed*" (interviewee 23).

### 4.4. Conflict of Values: Science Is Not a Business

The "engine" of most blockchain startups is the system of tokens, which motivate scientists, are used for voting and, ultimately, attract investors. At the heart of Bitcoin, the most successful blockchain project, as envisaged by its architect Satoshi Nakamoto, lies an incentive system. Bitcoin arranges incentives and rewards for all members of its ecosystem (miners, users, developers) in such a way that the output is stable, secure, and, at the same time, decentralized digital currency. It is through incentives that Satoshi Nakamoto has ensured that behavior beneficial to the common good of the system is encouraged and that harmful behavior is blocked [62].

Incentive design principles were conceptualized as one of the key benefits of blockchain. They entered other cryptocurrencies and then got integrated into more complex and diverse projects. The developers lay down what behavior will be encouraged by the participants. The mechanism of encouragement itself is material/financial: through tokens, which ultimately turn into money. "The blockchain community understands that blockchains can help align incentives among a tribe of token holders. Each token holder has skin in the game. But the benefit is actually more general than simply aligning incentives: you can design incentives of your choosing, by giving them block rewards. Put another way: you can get people to do stuff, by rewarding them with tokens. Blockchains are incentive machines" [63].

However, this "tokenization" of science has met mixed reactions on the part of the interviewees. For scientists, the desire for recognition and the pursuit of truth—non-monetary incentives—are arguably no less important than material ones [64]. The introduction of quantitative metrics and market mechanisms might corrupt science as a social institution: "*when you introduce monetary incentives into Wikipedia or peer review, you destroy them*" (interviewee 9). Besides, market logic (when everyone strives to maximize their profits) atomizes the scientific community, further undermining the logic of science as the collective search for the truth, when the common goal is more important than individual career success—the Mertonian norm of disinterestedness [65]. There is a dilemma: blockchain for science initiatives intend to build a self-regulating system run by scientists themselves, stimulating scientific progress in a self-governing sphere, but this new vision is based on a race for rewards and monetary incentives.

The advantages of the crypto economy (new funding opportunities, more freedom) are partially offset by the reluctance of scientists to take on the role of entrepreneurs, seek profits, and "sell" their research. Not every researcher is ready to act as an investor or a startup manager who attracts investments for their project, "*What these startups propose is a game . . . And many people don't like to play games. But the responsibility of a scientist is not to play games or to participate in community life, but to do research. To solve problems. And not to sell your research to others*" (focus group discussion 1).

### 4.5. Distrust of Politics in DAOs and On-Chain Voting

However, it was the political dimension of blockchain for science (DAOs, peer-to-peer funding, on-chain governance) that elicited the strongest criticism. The participants argued that public decision-making is allegedly prone to vote manipulation, collusions, and mob rule. Furthermore, open discussions and on-chain voting would mean that only fashionable and popular projects would get support. More marginal and innovative approaches would be even further marginalized and pushed aside, "*This is pretty similar to likes on Facebook. The one who gets more likes would succeed, regardless of the quality of their research. For example, I need to get support for my project on the DAO. And I just write to all my friends—guys, put a like to my project. Just as I ask them to upvote the video of my daughter on YouTube, or a photo of my cat. They'll support me because I asked them, not because my project is good*" (focus group discussion 1).

Another argument against self-regulation of science, DAOs, and p2p funding implies that, in academia, competition is much more powerful than cooperation, scientists exist in a Hobbesian state of perpetual war [66], and any attempt of e-democracy in science, through DAOs or any other digital tools, would only exacerbate existing conflicts and stall the research:

> "*Moderator: If scientists themselves would set the rules and allocate funds, would it work out? —No. Because it means war. Nobody wants a war. Somebody from the outside should make decisions—officials, administrators. They are more impartial*" (focus group discussion 4).

In fact, the very values of visibility and transparency—the strong points of blockchain and open science [11]—are not actually welcomed among scientists. It turned out that not everybody supports these values and it is ready to make their practices and decisions (as a researcher, a reviewer, an editor, a board member) visible, "*It's like the Internet, when we began to use it. All our conflicts moved online. Visibility creates risks. Especially if you make voting and decision-making public. All the conflicts between different schools, cliques, research groups would flare up in the open*" (focus group discussion 4).

Science, as an institution, functions as a network. This a very powerful metaphor, which is shared by scholars themselves, the historians of science and STS (the concept of "invisible colleges" [7]), public and corporate experts, IT companies, etc. However, this study has pinpointed a major discrepancy. For DAO and blockchain proponents, the network of scientists is flat, homogeneous, it consists of independent individuals making decisions, and communicating across borders (akin to nodes and miners in Bitcoin). However, the scientists see this network as a grouping of cliques/schools/gangs, collective entities with their conflicting values and interests. Being powerful, these groupings would reproduce themselves in any environment.

> "*Speaker #1: There would not be much discussion.*
> *Speaker #2: The strongest faction would emerge and then crush the others during voting.*
> *Speaker #3: Not the strongest, but the most insolent one.*
> *Speaker #1: Corrupted coalitions would emerge. Open discussions, tokens, voting on the blockchain would not make the rules of the game more fair and transparent. Same mafias would take control*"
> (focus group discussion 4).

### 4.6. Abuse of Power: Who Owns the Blockchain?

Finally, the interviewees expressed concern regarding the (invisible) power structures behind a blockchain system, and a power imbalance between the developers and ordinary users. Corporate applications aside, DLT itself, as a technology, is arguably not so decentralized and libertarian as its creators and evangelists have portrayed it—blockchain coders and developers enjoy an advantage over users, effectually acting as sovereign authority of their platforms [67] (p. 156). As one interviewee put it, "*Buterin [co-founder of Ethereum] preaches decentralization, but the meta-platform which sets the basic rules, the infrastructure on which a whole lot of independent projects and tokens is based, belongs to him. Decentralized autonomous organizations (DAOs) on Ethereum are not truly independent because they*

*are founded on his code. The more numerous and independent these projects become, the more his monopoly is entrenched"* (interviewee 15).

The issue of the ownership of data was discussed frequently: *"In the ideal world, nobody owns the blockchain. It is a distributed registry. But in the real world, a blockchain belongs to the one who invested in it, in the code, in the data management . . . Some countries, the USA, China, do really cool things, and one day they would offer our scientists to use DLT tools they've developed. A helping hand, so to speak. But this helping hand might as well become a Trojan horse, a tool to recruit our scholars and to get access to their data. So it is a must for blockchain initiatives to be decentralized and secure to the uttermost"* (interviewee 22).

If blockchain technologies are implemented from above, by the states that are eager to make use of transparency and data permanence that DLT offers, the scientists would face the perspective of increased control and coercion—an extra obligation to register all of their data on the blockchain, for instance. Proponents of new technology hail this as an essential step towards better science [8] (p. 11), but, from an end-user perspective, this could look more like a burden than an advantage.

## 5. Discussion

The barriers and challenges that are outlined in Section four do not necessarily imply that blockchain applications have no future in academia. However, these approaches have to be adopted by the academic community and, even before that, discussed and tested, in order to get beyond being just crazy ideas. Therefore, it is worth suggesting that the academic community's needs and concerns are investigated further (as blockchain gains traction) in more wide-ranging quantitative studies.

In the meantime, corporate players are beginning to take an interest in blockchain in science [34,68]. This trend reflects the general dynamics of blockchain's fortunes: creators and enthusiasts of technology conceived it as a path to digital democracy, life without banks, states, and corporations, whereas DLT are becoming a tool to optimize management processes and strengthen the power of the states and private companies. "Corporations enjoy a significant head start in the race to program their logics into mainstream blockchain applications, as well as the capacity to enact state policies that block new applications threatening future disintermediation . . . the corporatization of blockchain toward the ends of corporate sovereignty" [67] (p. 157).

Additionally, there are trends more favorable towards the implementation of blockchain in the management of science. In recent years, scholars all over the world have been increasingly mastering cloud applications aimed at automating different stages of the research cycle: note-taking (Evernote), collaborative writing (Authorea, GoogleDocs, Overleaf), references and citations management (Mendeley, Zotero), and data exchange (Figshare, GitHub). Experimental approaches have permeated the practice of science, be it new publishing models (open access), new metrics, such as altmetrics, new forms and practices of reviewing (open peer review, collaborative peer review). Blockchain fits into this trend quite naturally.

The problem is that existing DLT projects for science lack a "killer app", simple and efficient, aimed at solving one or maximum two problems evident to thousands of scientists—an approach modeling the success of the applications that are mentioned above. Projects under development lean toward complex solutions that reshape the rules of the game in the whole scientific ecosystem. Such an ambitious approach is riskier, and it would succeed only if enough stakeholders throw their weight behind a specific project. "Introducing a blockchain for research and its successful adoption will depend on the collaboration between all stakeholders: funders, government, institutions, publishers, and researchers themselves, whether in their role as researcher, reviewer, editor or author" [8] (p. 15).

One potential path for such a collaboration could draw on an example of an altogether different sphere: stolen bikes in the Netherlands. IBM has made a certain Hyperledger-based blockchain (BikeBlockchain) and a smartphone app: once a bike is stolen, the app sends relevant information to the police and insurance companies that cover the thefts. In this use, case blockchain brings different agencies (cyclists, the police, insurance companies) together, reducing administrative overhead and time expenditures (bicycle theft reporting is made easier). Yet, it was not the benefits of blockchain

per se (a decentralized data base open to independent parties without opening up their data), but the initiative and vigour of IBM, a private company that invested money, time, and efforts to bring all stakeholders together (this case is reviewed in [21]). Accordingly, probably a successful use case for DLT in science might resemble this example: an obvious problem to be solved, and a highly motivated intermediary with enough resources to bring different parties together.

This case also gives an instructive example of blockchain technology acting as a driver for open innovation, defined as "a distributed innovation process based on purposively managed knowledge flows across organizational boundaries" [69] (p. 17). Furthermore, blockchain lowers the cost of innovation by reducing the transaction costs and facilitating cooperation between parties. Blockchain institutional innovations might foster new forms of cooperation and policy coordination [17].

One could argue that blockchain has every chance of creating new opportunities and opening new avenues in the management of science. However, mere promises of a brighter digital future are not enough. To make it to the top, DLT projects in science should succeed in reaching out to individual scientists, as well as to the scores of fragmented academic "tribes" [70]. Successful applications should seamlessly integrate into scientists' daily lives, into existing practices and procedures, should make scholars' work more comfortable rather than more demanding. Finally, they should prove to key stakeholders, especially the universities and major foundations, that blockchain provides a standard of accuracy, transparency, and reliability.

**Funding:** This research was funded by the Russian Foundation for Basic Research (RFBR), grant number 18-29-16184.

**Conflicts of Interest:** The author declares no conflict of interest. The funder had no role in the design of the study; in the collection, analyses, or interpretation of data; in the writing of the manuscript, or in the decision to publish the results.

## Appendix A. List of Informants and Details on Focus Groups

Interviewee 1, developer & founder, academic publishing platform JES.su

Interviewee 2, co-founder, DEIP blockchain startup

Interviewee 3, founder, Agrello digital identity startup

Interviewee 4, chairman, International Society of Blockchain for Science

Interviewee 5, researcher, Research Institute for Cryptoeconomics (University of Vienna)

Interviewee 6, researcher, Blockchain & Society Policy Research Lab, Institute for Information Law, University of Amsterdam

Interviewee 7, researcher, Blockchain & Society Policy Research Lab, Institute for Information Law, University of Amsterdam

Interviewee 8, founder and investor, Expload

Interviewee 9, librarian, TechnischeInformationsbibliotek (Hannover, Germany)

Interviewee 10, cofounder, ARTiFACTS

Interviewee 11, founder, Melda.io

Interviewee 12, developer, DaMaHubDecentralised Data Infrastructure for Science.

Interviewee 13, cofounder, bloxberg: Blockchain Infrastructure for Scientific Research

Interviewee 14, cofounder, Molecule.to

Interviewee 15, researcher, State Academic University for the Humanities (GAUGN), Moscow, Russia.

Interviewee 16, developer, ipuniversity.ru

Interviewee 17, researcher, Department of Computer Modelling and Multiprocessor Systems, Saint Petersburg State University, Russia

Interviewee 18, vice-principal, Novosibirsk State University, Russia

Interviewee 19, dean, Faculty of Infocommunication Technologies, ITMO University, Russia

Interviewee 20, head, School of Data Economy, Far Eastern Federal University, Russia

Interviewee 21, researcher, Saint Petersburg Polytechnical University, Russia

Interviewee 22, vice-principal, Moscow Technological Institute, Russia

Interviewee 23, developer responsible for academic degree management, Penza State University, Russia

Interviewee 24, founder, Credentia.ru

Focus group discussion 1, Institute of Linguistics, Russian Academy of Sciences

Focus group discussion 2, Department of History, Chelyabinsk State University, Russia

Focus group discussion 3, Institute of Non-Organic Chemistry, Russian Academy of Sciences

Focus group discussion 4, Saint Petersburg Electrotechnical University, Russia

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
