# Peer review of "Blockchain Adoption in Academia: Promises and Challenges"

_2199-8531, doi:10.3390/joitmc6040117_

Round 1

Reviewer 1 Report

In this paper, the authors consider an overview of the opinions of researchers on the implementation and problems of the use and development of blockchain technology in the scientific environment. Interesting and topical problems are considered (use of actual sources, organization of financial reporting on competitions and grants, etc.), examples of solutions are described. What is the contribution of the paper and what problem is it trying to solve? The main contribution of the article should be mentioned. I would like to see positive examples of blockchain implementation in academia / universities / laboratories. The paper should also indicate whether the interviewees are experts in this field.

The introduction should be supplemented with references to works that carried out such research, if any.

Line 47-56. Disconnected paragraph. At the beginning of the paragraph, it is said about the need for research, but then the topic abruptly turns to comparison. It is probably better to correct the paragraph and tell first why you need to conduct research, and then make a comparison.

Line 80: The authors summarize that the studies were conducted in 2018-2019. Since the blockchain is developing very rapidly, perhaps it should be supplemented with more recent opinions? It is also possible, against the backdrop of the pandemic, the community reconsidered the importance of this technology.

The text lacks links to some sources, including in lines 122-124.
Line 122-124: Is it true that the first study on this topic appeared only in 2016? There is a lack of information and description of this study.

Line 154-171: Is blockchain really necessary to facilitate and speed up publishing and peer review?
The article mentions that this technology will make it possible to share information with other researchers, but already now many publications practice the publication of preprints (including in this journal).

Line 172-205: This solution would really be convenient for many researchers and make it easier for them to organize financial reporting. However, this will require other participants to switch to this technology, who may refuse to perform additional procedures.

Paragraph 4: Here the authors list the opinions of scientists on the implementation and use of this technology. In many interviews, however, the authors do not offer solutions to these implementation constraints. It may be worth expanding this paragraph with suggestions from the authors.

Reviewer 2 Report

The paper represent comprehensive research on the usage of DLT technologies in academia. The paper needs some minor corrections.  

Line 5: Blockchain has received much attention recently, due to its promises of verifiable, permanent, decentralized, and efficient data handling.

Blockchain does not efficiently handle data. Same data is duplicated among countless peers, for every new data entry an consensus must be reached that consumes a lot of energy in case of PoW and other factors. Please provide some basics for this claim

Line 36: The appeal of blockchain to industry and academia builds upon the promise to make data stable, immutable, transparent, and decentralized.

What does stable data mean?

Round 2

Reviewer 1 Report

The authors have reviewed my previous comments, I have no further comments. I thank the authors for the quick elimination of comments and wish you success in your research.

Reviewer 2 Report

I do not have any further suggestions